# Central nervous system infection in the intensive care unit: Development and validation of a multi-parameter diagnostic prediction tool to identify suspected patients

Hugo Boechat Andrade[1,2]*, Ivan Rocha Ferreira da Silva[3], Justin Lee Sim[3], José Henrique Mello-Neto[1], Pedro Henrique Nascimento Theodoro[1], Mayara Secco Torres da Silva[1], Margareth Catoia Varela[4], Grazielle Viana Ramos[5], Aline Ramos da Silva[5], Fernando Augusto Bozza[1,5], Jesus Soares[6], Ermias D. Belay[6], James J. Sejvar[6], José Cerbino-Neto[4], André Miguel Japiassú[1]

1 Intensive Care Unit, Evandro Chagas National Institute of Infectious Diseases (INI), Oswaldo Cruz Foundation (Fiocruz), Rio de Janeiro, RJ, Brazil, 2 Sexually Transmitted Diseases Sector, Biomedical Institute, Universidade Federal Fluminense (UFF), Niterói, RJ, Brazil, 3 Department of Neurological Sciences, Rush University Medical Center, Chicago, IL, United States of America, 4 Immunization and Health Surveillance Research Laboratory, Evandro Chagas National Institute of Infectious Diseases (INI), Oswaldo Cruz Foundation (Fiocruz), Rio de Janeiro, RJ, Brazil, 5 Department of Critical Care, D'Or Institute for Research and Education, Rio de Janeiro, RJ, Brazil, 6 Division of High-Consequence Pathology and Pathogens, National Center for Emerging and Zoonotic Infectious Diseases, Centers for Disease Control and Prevention, Atlanta, GA, United States of America

* hugo.boechat@ini.fiocruz.br

## Abstract

### Background

Central nervous system infections (CNSI) are diseases with high morbidity and mortality, and their diagnosis in the intensive care environment can be challenging. *Objective*: To develop and validate a diagnostic model to quickly screen intensive care patients with suspected CNSI using readily available clinical data.

### Methods

*Derivation cohort*: 783 patients admitted to an infectious diseases intensive care unit (ICU) in Oswaldo Cruz Foundation, Rio de Janeiro RJ, Brazil, for any reason, between 01/01/2012 and 06/30/2019, with a prevalence of 97 (12.4%) CNSI cases. *Validation cohort 1*: 163 patients prospectively collected, between 07/01/2019 and 07/01/2020, from the same ICU, with 15 (9.2%) CNSI cases. *Validation cohort 2*: 7,270 patients with 88 CNSI (1.21%) admitted to a neuro ICU in Chicago, IL, USA between 01/01/2014 and 06/30/2019. *Prediction model*: Multivariate logistic regression analysis was performed to construct the model, and Receiver Operating Characteristic (ROC) curve analysis was used for model validation. Eight predictors—age <56 years old, cerebrospinal fluid white blood cell count >2 cells/mm$^3$, fever (≥38˚C/100.4˚F), focal neurologic deficit, Glasgow Coma Scale <14 points, AIDS/HIV, and seizure—were included in the development diagnostic model (P<0.05).

**Data Availability Statement:** All relevant data are within the paper and its Supporting Information files.

**Funding:** The author(s) received no specific funding for this work.

**Competing interests:** The authors have declared that no competing interests exist.

## Results

The pool data's model had an Area Under the Receiver Operating Characteristics (AUC) curve of 0.892 (95% confidence interval 0.864–0.921, P<0.0001).

## Conclusions

A promising and straightforward screening tool for central nervous system infections, with few and readily available clinical variables, was developed and had good accuracy, with internal and external validity.

## Introduction

Infectious diseases with significant public health impact due to potential severity, such as encephalitis and hemorrhagic fever, are a challenge for health systems and health authorities worldwide [1]. Thus, Intensive Care Units (ICUs) can be an essential target for establishing sentinel syndromic surveillance, optimizing resources by precisely focusing on new diseases with tremendous potential for severe morbidity and mortality [2].

Among undiagnosed severe infectious illnesses, encephalitis may be considered a hallmark disease [3]. It is a severe clinical manifestation associated with many autoimmune and infectious diseases, including recently identified emerging and reemerging pathogens [4–6]. Besides, its true incidence is difficult to determine because many cases are unreported, the diagnosis may not be considered, or a specific infectious etiology may never be confirmed [6–8].

Robertson et al. [9] conducted a systematic literature review and meta-analysis of 154 studies of Central Nervous System Infections (CNSI) published between 1990 and 2016, 71 of them with incidence data. A total sample size of 130,681,681 individuals with 508,078 cases across all studies was included, with a global prevalence of 0.4%.

The encephalitis incidence varies from 3.5 to 7.4/100,000 patient-years, and it occurs worldwide. Some etiologies have a global distribution (e.g., herpesviruses), while others are geographically restricted (e.g., arboviruses) [4, 10]. Other CNSI, including meningitis and brain abscesses, are less rare: hospitalization and ICU admission rates varied from 1 to 4.5%. The incidence of brain abscess is approximately 8% of intracranial masses in developing countries and 1% to 2% in the Western countries, with around four cases occurring per million [11–14].

It is more challenging to generalize for encephalitis, as few population-based studies exist. Many possible pathogens are implicated, and most cases are not reported to health authorities. Still, in most cases, a cause is never found [15].

We aimed to develop and validate a diagnostic model that allows for the quick screening of patients suspected of having CNSI, consequently, encephalitis, using a readily available clinical dataset. Its simplicity could enable application on an individual level and potential for population screening and even large databases. A neurological diagnostic prediction model for delirium in adult ICU patients [16] was previously developed. Still, the model described in the present article, to our knowledge, is the first model intended for monitoring CNSI in ICUs.

## Materials and methods

This multivariate diagnostic model was developed and validated following the Transparent Reporting of a Multivariable Prediction Model for Individual Prognosis or Diagnosis (TRIPOD) statement [17]. The checklist is on the **S1 File.**

## Ethics statement

The study was approved by the local Institutional Review Board (CAAE 16876819.9.0000.5262), which waived the need for informed consent, as the data were analyzed anonymously. No interventions were carried out, and data collection was not burdensome to patients. This report's findings and conclusions are those of the authors and do not necessarily represent the Centers for Disease Control and Prevention's official position.

## Data collection and potential predictive variables

We first performed an observational retrospective cohort study of patients admitted between January 1st, 2012, and June 30th, 2019, in the 4-bed ICU of a 25-bed hospital located at Evandro Chagas National Institute of Infectious Diseases (INI), Oswaldo Cruz Foundation (Fiocruz), Rio de Janeiro, Brazil.

We reviewed the medical records of all 869 consecutive patients admitted, for any reason, excluding readmissions (80) to ICU during the period of data collection and patients (6) with critical missing data in medical records. So, 783 patients were included in the development cohort (DC).

Potential predictive variables selected were those known associated with CNSI, its severity, and outcome, and readily available in emergency departments or ICUs [18] to calculate predictive score systems, like Simplified Acute Physiology Score (SAPS) 3 [19]. The following were collected in the first 24 h of ICU admission:

1. Age, sex, dates of hospital and ICU admission and discharge, the patient outcome at discharge (alive/dead).

2. Clinical and laboratory data: SAPS 3 and Sequential Organ Failure Assessment (SOFA) [20] prognostic scores and the lowest Glasgow Coma Scale (GCS) [21]; fever $\geq$ 38˚C (100.4˚F) within the 72h before or after the presentation, AIDS/HIV (Acquired Immunodeficiency Syndrome / Human Immunodeficiency Virus) infection status.

3. Neurologic signs/symptoms: cerebrospinal fluid (CSF) white blood cell count (WBC)/mm$^3$, and those syndromes defined by the SAPS 3 score [22] and Venkatesan et al.: *Encephalopathy*—altered mental status, defined as decreased or altered level of consciousness/vigilance disturbances, confusion, disorientation, behavioral changes, or other cognitive impairment, lasting $\geq$24 h with no alternative cause identified; new onset of *focal neurologic signs* (hemiplegia, paraplegia, tetraplegia); generalized or partial *seizures* not entirely attributable to a preexisting seizure disorder.

When missing values were less than 20%, imputation for missing variables was considered. We used logistic regression to impute binary variables and predictive mean matching to impute numeric features.

## Outcomes

Central nervous system infection was defined as any case of the following diseases, diagnosed between 48 h before and five days after ICU admission:

- Cerebral abscess or suppurative intracranial infections: Symptoms of a mass lesion, seizures, signs of focal deficit, and cerebral lesion documented by neuroimaging (magnetic resonance imaging/computed tomography) or anatomical evidence.

- Encephalitis: Involvement of the brain parenchyma by infectious agent inducing neurological symptoms. It could be documented by CSF abnormalities, serology, isolation of the

causal agent, neuroimaging. The criteria for encephalitis diagnosis were those defined by Venkatesan et al. and shown on S1 Table in the **S2 File**.

- Meningitis: Patients without criteria for encephalitis, but with symptoms of the meningeal syndrome (headache, fever, irritability, and stiff neck, with or without focal neurological signs) with positive CSF culture or CSF abnormalities compatible with meningitis, serology, isolation of the causal agent, neuroimaging.

Two physicians (HBA and JHN) independently reviewed the medical records. The diagnosis of CNSI was considered if it met at least two of the following criteria: clinical syndrome, neuroimaging, CSF analysis, and microbiological exams (blood and CSF cultures, serologies). All patients were submitted to computed tomography. One-third of the DC and VC1 could not be submitted to lumbar puncture because of formal contraindications for the procedure (all of them with brain abscesses). Those with laboratory diagnosis of CNSI but no symptoms were classified as asymptomatic CNSI.

## Statistical analysis

Statistical analyses were performed, and figures created using the MedCalc® application, version 19.3, for Microsoft Windows®. Categorical variables were expressed as the absolute numbers and percentages in each category. Chi-square and Fisher's exact tests were used to analyze categorical variables. Continuous variables were expressed as medians with interquartile ranges (IQR) and analyzed by Mann–Whitney U-test. A p-value <0.05 and 95% confidence interval (CI) indicated significance for all tests.

## Predictor selection and model construction

Sixteen variables were analyzed, and those associated (p<0.05) with the outcome were included in a Least Absolute Shrinkage and Selection Operator (LASSO) regression to minimize the potential collinearity of variables, as shown on S2-S4 Tables in **S2 File**. This approach refined and defined the final multivariate logistic regression model, avoiding collinearity [23].

Values were missing in the DC for body temperature (1%), encephalopathy (2%), and the Glasgow Coma Scale score (1%). Data for all other variables were complete. The optimal cutoff point, where Youden's index is maximum, converted continuous to categorical data before regression. Subsequently, variables identified by LASSO regression analysis were entered into multivariate logistic regression models, and those that were statistically significant were used to construct the diagnostic model.

We used bootstrapping techniques to adjust for overly optimistic estimates of the predictors' regression coefficients in the final model (overfitting): one thousand random bootstrap samples resulted in shrunken regression coefficients [24]. Finally, the calibration slopes of the regression lines for the cohorts updated the model.

## Assessment of accuracy

The model's potential ability to discriminate between patients with and without central nervous system infection was quantified by diagnostic accuracy measures, such as sensitivity, specificity, predictive values, likelihood ratios, and the area under the receiver-operator characteristic (ROC) curve (AUC). The sample size calculated for the AUC was at least 96 patients (12 positive cases and 84 negative cases) for the following parameters: alpha 0.05, beta 0.2, minimal AUC of 0.9, null hypothesis 0.5 (no discriminating power).

## Model validation

To validate the generalizability of the algorithm, we used two cohorts:

- *Internal validation cohort (VC1)*: 163 patients with 15 (9.2%) cases of CNSI were included. One case (6.6%) was classified as asymptomatic. One hundred seventy-seven patients were reviewed for a prospective cohort study in INI between July 1st, 2019, and July 1st, 2020. Ten readmissions and four records were excluded because of critical missing data in medical records. Data for all variables were complete.

- *External validation cohort (VC2)*: 7,270 patients with 88 (1.2%) cases of CNSI were included. 18 (20.45%) of the CNSI cases were classified as asymptomatic. A retrospective cohort of patients admitted between January 1st, 2014, and June 30th, 2019, in the neuro ICU from Rush University Medical Center, Chicago, IL, USA.

The variables required for calculating the VC2 were collected and crosschecked by two of the authors (IRFDS and JLS). A case of CNSI was established by the same criteria as described herein for the development cohort. The data about AIDS/HIV was missing for 40% of the patients. All the other variables had less than 20% of missing data.

## Results

### Characteristics and outcomes from the development and validation cohorts

Table 1 summarizes the demographic and clinical characteristics of the DC, VC1, and VC2. The detailed profile of the 783 patients from the DC, with 97 (12.38%) cases of CNSI and 9 (9.28%) of them asymptomatic, as shown in S2 and S3 Tables in S2 File. The DC's variables associated with the outcome (p<0.05) were selected for the LASSO regression, as shown in S4 Table in S2 File. The S5 Table in S2 File shows the global microbiological profile of the cohorts.

There were some significant differences (P<0.05) between the DC and VC1: median age, the prevalence of AIDS/HIV, the median SOFA score, the ICU/hospital mortalities; but no significant difference (P>0.05) in CNSI/asymptomatic CNSI prevalence, or the median SAPS 3 score. Those differences can be explained from a clinical view: severe coronavirus disease 2019 (COVID-19), by SARS-CoV-2, was the reason for the admission to ICU of 61/163 patients (37.42%) from the VC1, without CSNI cases among them.

The VC2 was a completely different sample compared to the INI cohorts in all characteristics (p<0.05). The most remarkable differences between the DC and VC2 are the prevalence of CNSI, asymptomatic CNSI, AIDS/HIV, and surgical patients, which are expected for a neurointensive care unit.

**Diagnostic model, calibration slopes and recalibration.** Table 2 shows the results of the multiple logistic regression: AIDS/HIV, Age <56 years old, CSF WBC >2 cells/mm3, fever (body temperature ≥38˚C), focal neurological deficit, encephalopathy, GCS <14 points, and seizures were predictors independently associated with central nervous system infections diagnoses (p<0.05).

S1 Fig in S2 File shows the linear regression lines for each cohort and their calibration slopes. The model overestimated the risk of CNSI in the VC2 by about 40% more than DC's risk estimation. The coefficients of the individual predictors were updated to recalibrate the model [25]. Each predictor with a factor that is the estimated calibration slope (0.5981) and adding the estimate of α' (1,341; the intercept of the calibration slope model) to the original intercept, adjusted to the local prevalence of the disease as an additional correction coefficient of 0.1 [26, 27].

**Table 1. Demographic and clinical characteristics of the development (DC) and validation cohorts (VC1 and VC2).**

| Variables | VC 1 (n = 163) | CD | U / $\chi^2$ | DC (n = 783) | CD | U / $\chi^2$ | VC 2 (n = 7,270) |
|---|---|---|---|---|---|---|---|
| AIDS/HIV | 52 (31.9%) | 22.8% | **<0.0001** | 428 (54.7%) | 53.7% | **<0.0001** | 73 (1%) |
| Age (years) | 57 (40–69 IQR) | 6 (3–9) | **0.0004** | 48 (37–61 IQR) | 10 (7–12) | **<0.0001** | 60.2 (45–72 IQR) |
| COVID-19 cases | 61 (37.42%) | 37.42% | **<0.0001** | 0 (0%) | 0 | 0.8999 | 0 (0%) |
| CSF WBC(/mm³) | 1 (0–15) | 0 (0–0) | 0.1882 | 0 (0–27 IQR) | 0 (0–0) | 0.0623 | 0 (0–10) |
| Fever | 13 (8%) | 4% | 0.1397 | 94 (12%) | 9,5% | **<0.0001** | 182 (2.5%) |
| Encephalopathy[a] | 37 (22.7%) | 15.1% | **0.0002** | 296 (37.8%) | 25% | **<0.0001** | 4580 (63%) |
| Focal Deficit[a] | 3 (1.8%) | 0.4% | 0.7896 | 17 (2.2%) | 18% | **<0.0001** | 1454 (20%) |
| GCS (points) | 15 (15–15 IQR) | 0 (0–0) | **0.0004** | 15 (13–15 IQR) | 4 (1–6) | **<0.0001** | 10 (7–14 IQR) |
| Hospital death | 72 (44.2%) | 10.2% | **0.0135** | 266 (34.0%) | 21.41% | **<0.0001** | 915 (12.59%) |
| ICU death | 62 (38.0%) | 11.7% | **0.0025** | 206 (26.3%) | 18.3% | **<0.0001** | 582 (8%) |
| LOS hospital (days) | 15 (8–24 IQR) | 0 (-3-2) | 0.6933 | 14 (7–31.75 IQR) | 7 (3–11) | **<0.0001** | 7.06 (5–12 IQR) |
| LOS ICU (days) | 9 (5–17 IQR) | 2 (1–3) | **<0.0001** | 6 (3–12 IQR) | 3 (1–4) | **<0.0001** | 3 (1–5 IQR) |
| SAPS 3 (points) | 56 (48–63.75 IQR) | -1 (-3-2) | 0.5358 | 56 (47–67 IQR) | 10 (6–12) | **<0.0001** | 45 (36–51 IQR) |
| Seizures[a] | 9 (5.5%) | 0 | 0.9879 | 43 (5.5%) | 9.5% | **<0.0001** | 1091 (15%) |
| Sex (male) | 97 (59.5%) | 1.2% | 0.8107 | 458 (58.5%) | 8.2% | **<0.0001** | 3657 (50.3%) |
| SOFA (points) | 6 (4–9 IQR) | 1 (0–2) | **0.0144** | 5 (2–9 IQR) | 1 (0–3) | **0.0255** | 4 (1–6 IQR) |
| Surgical patients | 6 (3.7%) | 0.5% | **0.7107** | 25 (3.2%) | 21.5% | **<0.0001** | 1793 (24.66%) |
| Central nervous system infections (CNSI) | 15 (9.2%) | 3.19% | 0.2523 | 97 (12.39%) | 11.2% | **<0.0001** | 88 (1.2%) |
| Asymptomatic CNSI (% of cases) | 1 (6.6%) | 2.68% | 0.0978 | 9 (9.28%) | 11.17% | **<0.0001** | 18 (20.45%) |

[a]Neurological reasons for intensive care unit (ICU) admission, from SAPS (Simplified Acute Physiology Score) 3. Bold: p<0.05. DC: development cohort. VC1: internal validation cohort. VC2: external validation cohort. CD: validation cohorts' differences from DC. IQR: interquartile range. COVID-19: Coronavirus disease 2019 by SARS-CoV-2. CSF: cerebrospinal fluid. WBC: white blood cell count/mm³. SOFA: sequential organ failure assessment score. GCS: lowest Glasgow Coma Scale in the first 24 h of ICU admission. Aids: acquired immune deficiency syndrome. Encephalopathy: any altered consciousness/vigilance disturbances—coma, stupor, obtundation, or delirium. Fever: temperature equal to or above 38 degrees Celsius or 100.4 degrees Fahrenheit. Focal deficit: hemiplegia, paraplegia, tetraplegia.

**Table 2. Multiple logistic regression final model and calibrated regression coefficient derived from the development cohort (DC).**

| Predictors | Odds ratio | 95% CI | Regression coefficients | P | Updated regression coefficients* |
|---|---|---|---|---|---|
| *Constant/Intercept* | | | *-5.741* | *0.001* | *-4.4* |
| AIDS/HIV | 3.127 | 1.702 to 5.745 | 0.455 | 0.003 | 0.273 |
| Age <56 (years) | 6.373 | 2.146 to 18.925 | 1.629 | 0.001 | 0.977 |
| CSF WBC >2 cells/mm³ | 140.42 | 23.027 to 856.281 | 4.322 | 0.001 | 2.701 |
| Encephalopathy | 3.100 | 1.395 to 6.893 | 1.098 | 0.005 | 0.658 |
| FEVER | 2.762 | 1.225 to 6.226 | 1.032 | 0.015 | 0.619 |
| Focal Neurologic Deficit | 16.206 | 4.140 to 63.433 | 2.554 | 0.001 | 1.532 |
| GCS <14 (points) | 4.873 | 2.254 to 10.538 | 1.52 | 0.001 | 0.912 |
| Seizures | 4.684 | 1.951 to 11.248 | 1.495 | 0.005 | 0.897 |
| *Local prevalence of CNSI** | | | | *0.1* | |

Overall Model Fit: Null model -2 log-likelihood 586.608. Full model -2 log-likelihood 281.020. Chi-squared 305.588. DF 8. P<0.0001. Cox & Snell R2 0.3231. Nagelkerke R2 0.6129. Hosmer & Lemeshow test: Chi-squared 2.5490 DF 8. P = 0.959.

Neurological signs: consider new-onset neurological syndrome.

CNSI: central nervous system infection. CI: confidence interval. Fever: temperature equal to or above 38 degrees Celsius or 100.4 degrees Fahrenheit. Encephalopathy: any altered consciousness/vigilance disturbances–coma, stupor, obtundation, or delirium. GCS: Glasgow coma scale. CSF: cerebrospinal fluid. WBC: white blood cell count/mm³. ICU: intensive care unit.

* Recalibration: updating intercept and coefficients according to the local prevalence of CNSI.

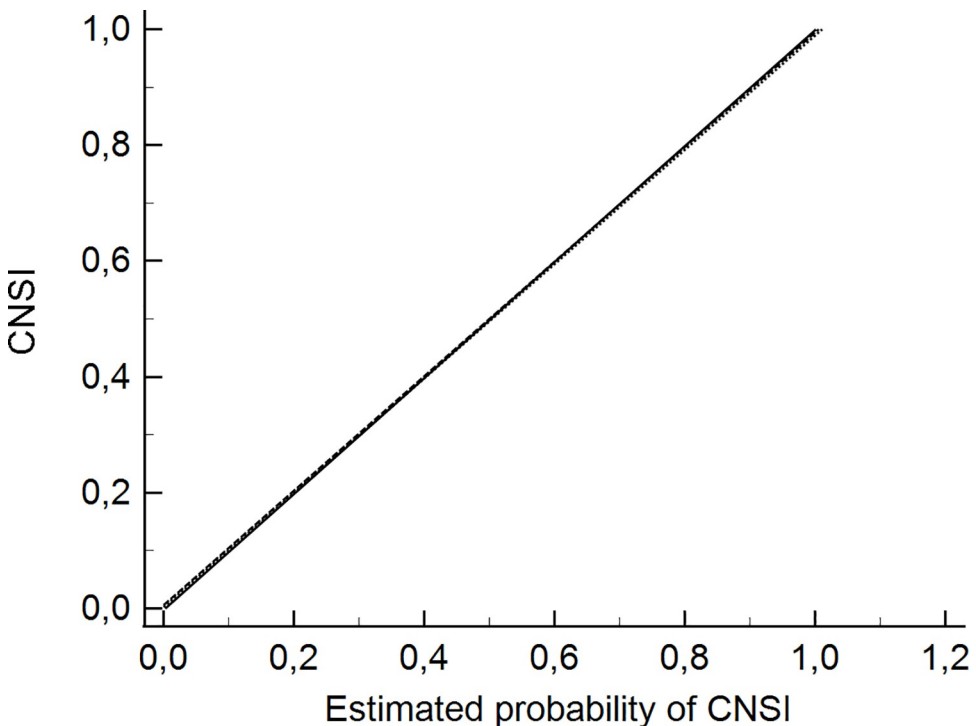

**Fig 1. Final updated calibration slope for the DC, VC1 and VC2.** DC: development cohort. VC1: internal validation cohort. VC2: external validation cohort. Solid line DC: y = -0.002583 (-0.02065 to 0.01548 CI; P = 0.7790) + 1.0013 x (0.9343 to 1.0683; P<0.0001) x. Dashed line VC1: y = 0.005774 (-0.02433 to 0.03588; P = 0.7054) + 0.9887 (0.8655 to 1.1119; P<0.0001) x. Dotted line VC2: y = 0.002511 (-0.02846 to 0.03348; P = 0.8735) + 0.9868 (0.8590 to 1.1146) x.

Fig 1 compares the regression lines after recalibration: there were no significant differences between the slopes (0.01257, P = 0.8790) or the intercepts (-0.007181, P = 0.7156) of DC vs VC1, nor between the slopes (-0.01443, P = 0.8339) or the intercepts (-0.003056, P = 0.8338) of DC vs VC2. The model regression equation was y = 0.00008489 (-0.01473 to 0.01490, P = 0.9910) + 0.9954 (0.9377 to 1.0531, P<0.0001) x, with a coefficient of determination $R^2$ of 0.42176 and the residual standard deviation of 0.2548. It suggested a well-calibrated final model.

**The formula for CNSI probability.** Estimated probability of central nervous system infection = 1 / [1 + exp—(-4.4 + 0.273 * "AIDS/HIV" + 0.9774 * "Age <56 years-old" + 0.6192 * "Fever (T≥38˚C)" + 0.6588 * "Encephalopathy" + 0.912 * "Glasgow Coma Scale <14 points" + 1.532 x "Neurologic Focal Deficit" + 0.897 * "Seizures" + 2.701 * "CSF WBC >2 cells/mm$^3$" + 0.1 * "Local CNSI prevalence in %")].

The presence or absence of a predictor was defined by 1/0 on the formula. The local prevalence must be entered as a percentage (e.g., 12%). If the local prevalence is unknown, the field can be left blank.

For example, an HIV-negative 70-years-old person with fever and encephalopathy in a low prevalence setting (1%) has a 4% probability of CNSI. On the other hand, a 40-year-old HIV patient with fever, hemiplegia, and seizures, in a high prevalence setting (10%), has a 71% risk of CNSI. The CNSI probability calculator can be tested on S3 File.

**ROC curve analysis.** Fig 2 compares the ROC curves for the development and the validation cohorts. The DC showed an AUC of 0.939 (CI 0.903 to 0.959, p<0.0001), while the VC1 an AUC of 0.978 (CI 0.945 to 0.994, p<0.0001), with a small but significant difference between areas: 0.0398, P<0.0192. The VC2 presented an AUC of 0.840 (CI 0.802–0.870, P<0.0001), with a significant difference (0.108, P<0.0004) when compared to DC's, expected for validation. The

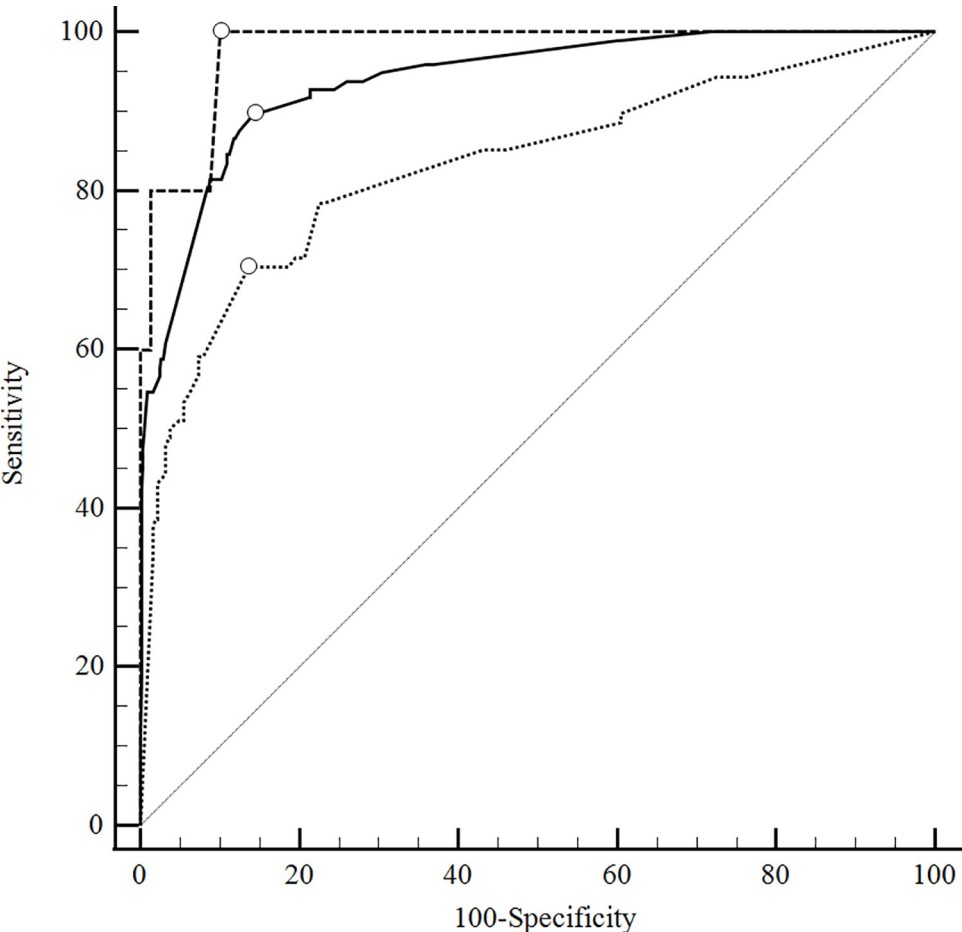

**Fig 2. ROC curves for DC, VC 1 and VC2.** AUC: area under the ROC curve. DC: development cohort. VC1: internal validation cohort. VC2: external validation cohort. ROC: Receiver operating characteristic. Solid line DC: AUC of 0.939 (CI 0.903 to 0.959, p<0.0001). Dashed line VC1: AUC of 0.978 (CI 0.945 to 0.994, p<0.0001). Dotted line VC2: AUC of 0.840 (CI 0.802–0.870, P<0.0001).

model adjustment did not change the ranking of the predicted risks, so the AUC was unaltered by the recalibration. The pool data's AUC was 0.892 (0.864–0.921, P<0.0001).

As a specialized hospital in infectious diseases, INI's AIDS/HIV prevalence was at least 100 times the Brazilian prevalence in the general population [28]. **Fig 3** shows a sensitivity analysis correcting for the importance of the HIV population in DC: HIV-negative patients had an AUC of 0.945 (CI 0.920–0.964, P<0.0001), not significantly different (0.0238, P = 0.4041) from HIV—positive patients' area under the curve (0.921 [CI 0.893 to 0.944, P<0.0001]).

**Measures of diagnostic accuracy of the model.**   **Table 3** shows the sensitivity, specificity, and likelihood ratios for each risk group on the model: low (0–10%), medium (possible CNSI, >10–50%), and high probability (probable CNSI, >50%). The optimal cutoff point was >0.1032 (10%), with a sensitivity of 88.69, a specificity of 85.57, a positive likelihood ratio of 6.21, and a negative likelihood ratio of 0.12.

## Discussion

We developed and validated a predictive model for aiding in diagnosing central nervous system infections in ICU patients. To our knowledge, this is the first of its kind for general

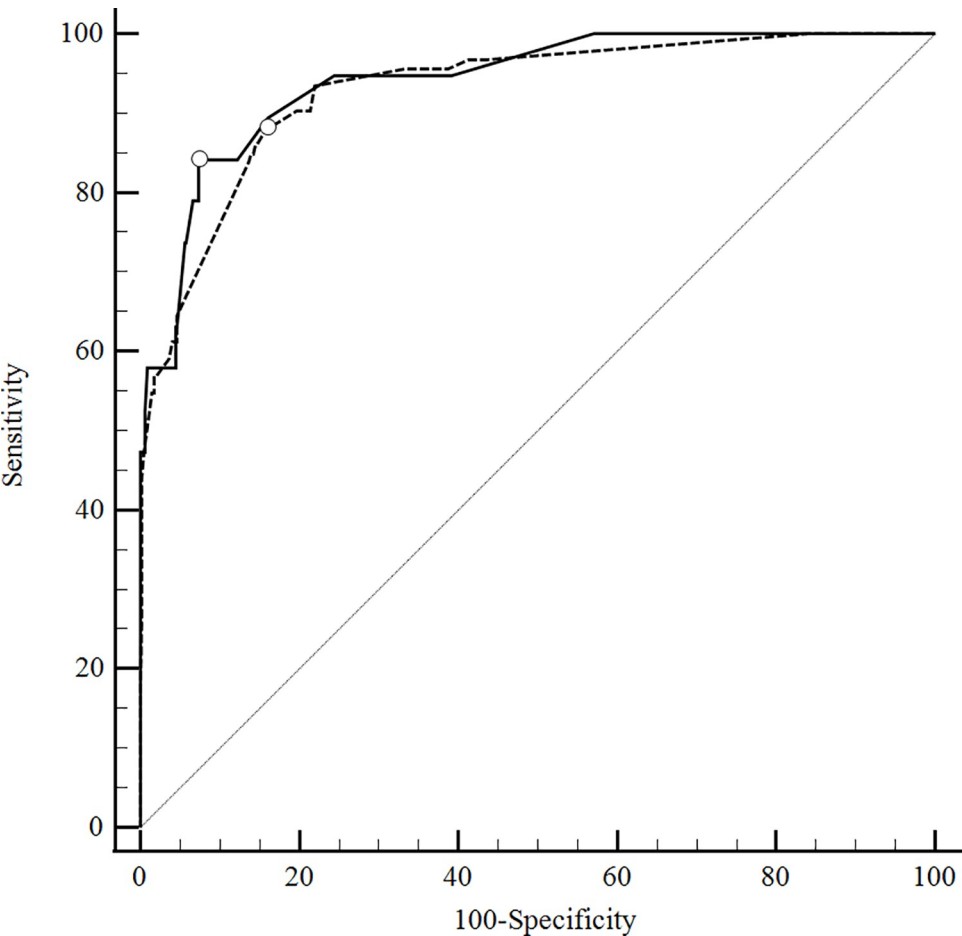

**Fig 3. ROC curves for HIV vs. non-HIV patients in DC.** AUC: area under the ROC curve. DC: development cohort. ROC: Receiver operating characteristic. HIV: human immunodeficiency virus. Solid line Non-HIV: AUC of 0.945 (CI 0.920–0.964, P<0.0001). Dashed line HIV: AUC of 0.921 (CI 0.893 to 0.944, P<0.0001). Difference between areas: 0.0238, P = 0.4041.

**Table 3. Measures of diagnostic accuracy infections, risk groups, and cutoff points of the ROC curve diagnostic model for central nervous system infections (CNSI).**

| Cut off | Low risk | | Medium risk—possible NSI | | | High risk—probable NSI | | |
|---|---|---|---|---|---|---|---|---|
| | 4% | 6% | 10%* | 15% | 35% | 50% | 60% | 80% |
| SEN | 95 (88–98.3) | 91.75 (84.4–96.4) | 89.69 (81.9–95) | 84.54 (76–91.1) | 57.73 (47–68) | 54.64 (44–65) | 46.39 (36–57) | 39 (25–50) |
| SPE | 69.53 (65.9–73) | 78.72 (75.5–81.7) | 85.57 (82.7–88) | 89 (86.5–91) | 97.5 (96–98.5) | 98.98 (98–99.6) | 99.71 (99–100) | 99.85 (99–100) |
| PLR | 3.11 (2.8–3.5) | 4.31 (3.7–5) | 6.21 (5.1–7.5) | 7.73 (6.1–9.7) | 23.3 (14–38.4) | 53.5 (25–114) | 162.66 (40–660) | 269 (27–1935) |
| NLR | 0.074 (0.03–0.2) | 0.1 (0.05–0.2) | 0.12 (0.07–0.2) | 0.17 (0.1–0.3) | 0.43 (0.3–0.5) | 0.46 (0.4–0.6) | 0.53 (0.4–0.6) | 0.6 (0.5–0.7) |
| +PV | 30.6 (28–33.2) | 37.9 (34.3–41.6) | 46.8 (42–51.6) | 52.2 (46.5–58) | 76.7 (67–84.4) | 88.3 (78–94) | 95.8 (85–99) | 97.4 (84–99) |
| -PV | 99 (98–99.6) | 98.5 (97–99) | 98.3 (97–99) | 97.5 (96–98.5) | 94.2 (93–95.4) | 94 (92.5–95.5) | 93 (92–94) | 92 (91–93) |

DC: 97 central nervous system infections / 783 patients.

*Youden index cut-off point associated criterion: >0.1032 (>10%). DC: development cohort. NLR: Negative Likelihood Ratio. PLR: Positive Likelihood Ratio. +PV: Positive Predictive Value. -PV: Negative Predictive Value. ROC: Receiver operating characteristic. SEN: Sensitivity. SPE: Specificity.

intensive care patients. Our model reliably predicted these infections based on seven readily available variables on admission. The clinical variables are widely used for the calculation of other prognostic scores, such as SAPS 3. The additional laboratory variable to help identify asymptomatic infections.

The DC and VC1 belonged to a referral center for infectious diseases, including AIDS/HIV, in the second-largest Brazilian urban center (Rio de Janeiro). That explains not only the prevalence of CNSI (12.4%), which is at least twice as high as in other Brazilian ICUs (1–5%) in general hospitals [13] but also that the prevalence of AIDS/HIV (54.7%) is 30-times as high as in Brazilian hospitals (1.8%) [28]. However, the prevalence of CNSI among our HIV-negative critical patients (5%) was like other general medical ICUs [28].

A CSF WBC count $\geq 5$ cells/mm$^3$ is one of the minor criteria for encephalitis diagnosis (S1 Table in S2 File). However, CSF may be devoid of cells in immunocompromised patients [29] or early in the course of infection [30], not excluding encephalitis. Therefore, the proposed algorithm adjusts the cutoff point to a more sensitive value of CSF WBC $>2$ cells/mm$^3$.

The microbiological profile is compatible with the current international literature: in a multicenter international study to understand the burden of community acquired CNSI, Erdem et al. showed that the most frequent pathogens were *Streptococcus pneumoniae* (n = 206; 8%) and *Mycobacterium tuberculosis* (n = 152; 5.9%). Cryptococci were leading pathogens in the subgroup of HIV-positive individuals. Ninety-six (8.9%) patients of INI's sample presented with clinical features of a subacute disease, suggestive of tuberculosis or neurosyphilis [31].

## Clinical relevance

Our findings suggest that the model may have great value in daily practice to help to screen patients with higher risks ($>10\%$), as we would catch 181/200 (90.5%) of CNSI cases, even with 28/200 (14%) of asymptomatic ones. The 19 cases classified as false negatives were the following:

- Postoperative subarachnoid or intracranial hemorrhage, with external ventricular drain and secondary infection—6 patients.

- Patients with missing data—5 patients.

- CNSI in a patient with previous neurological disease and poorly characterized new symptoms on the medical record, classified as asymptomatic CNSI—2 patients.

- Neurosyphilis—asymptomatic infection, ICU admission for other causes, LCR exam realized for distinct reasons—2 patients.

- Asymptomatic neurocryptococcosis—admitted for other infections, blood antigen exam was positive, so the patients were submitted to LCR analysis and neuroimage exam—4 patients.

The overall prevalence of SNI was 13.33%. In comparison, the prevalence estimated by the model would be 15% (in a low prevalence setting—1–2%) to 25% (in a high prevalence setting—10%), expected for a cutoff value (10%) for a model designed as a screening tool. The neurointensive ICU presented more false-positive cases, as neurological symptoms were more common.

Only 80/200 (40%) of all patients with CNSI had that suspicion on ICU admission. Hence, the initial suspicions are not reliable, even in specialized institutions. Among patients presenting to the emergency department at a single United States of America hospital with a clinical suspicion of meningitis who underwent lumbar puncture, the prevalence of meningitis (defined as cerebrospinal fluid white blood cell count $\geq 5$/mL) was 27%. In the broad spectrum

of adults with suspected meningitis, three classic meningeal signs (Kernig's sign, Brudzinski's sign, and nuchal rigidity) did not have diagnostic value [32], so better bedside diagnostic tools are needed.

An estimated probability of CNSI lower than 10% makes this hypothesis improbable, so the investigation of other diagnoses must be prioritized. On the other hand, a risk greater than 10% indicates that imaging exams and diagnostic lumbar puncture, if possible, should be considered. Finally, a chance greater than 50% suggests that complementary exams are mandatory or repeated if the diagnosis is unclear, and empirical treatment should also be considered.

The model's variables can be used for CNSI screening in large health system databases as well, provided the necessary variables are included, which could serve as a sentinel surveillance tool for encephalitis and other CNSI. Finally, it could also be used as a tool to calculate the pre-test probability of CNSI before other diagnostic tests, allowing earlier diagnosis and ensuring efficient use of research and diagnostic resources.

### Limitations of the study

This study has limitations. The completion of medical records data in research institutions might be better than in other institutions that are not research driven. That is why we chose to select as few variables as possible, present in almost every record in the database.

Retrospective studies have limitations and specific bias risks, so a prospective cohort was used for internal validation to reduce those biases. The high proportion of patients with AIDS/HIV, the high prevalence of CNSI, and the extremely low prevalence of surgical patients in the DC can influence the external validity of the tool, as well as its calibration. So, the VC2 was included to lessen those problems.

The calibration and the cutoff points should be validated in other scenarios, like emergency rooms, general/mixed ICUs, general wards, and even outpatients. The model showed worse performance with surgical patients and, naturally, with asymptomatic infections, as the diagnosis depends heavily on laboratory data. The use of CSF WBC count in the model lessens that limitation.

Encephalopathy and GCS are correlated variables, which could influence the accuracy of the model. However, both are commonly missing data in medical records. For that reason, SAPS 3 use both: the first as a more subjective criterion (quality of mental status) and the second as an objective one (quantitative measure of conscience). Besides, the LASSO regression and the bootstrapping did not recommend excluding one of them from the final model.

### Conclusions

A promising and straightforward screening tool for central nervous system infections, with few and readily available clinical variables, was developed and had good accuracy, with internal and external validity.

Future research is needed to validate this tool in other settings. It could provide a cost-effective means to successfully identify these cases and lead to more timely diagnostics and treatment in an intensive care setting.

### Supporting information

**S1 File. TRIPOD checklist.**
(DOCX)

**S2 File. Supporting data for "central nervous system infection in the intensive care unit: Development of a multi-parameter diagnostic prediction tool to identify suspected**

patients".
(DOCX)

**S3 File. Central nervous system infection probability calculator.**
(XLSX)

**S4 File. INI development cohort dataset.**
(XLSX)

## Author Contributions

**Conceptualization:** Hugo Boechat Andrade, Fernando Augusto Bozza, James J. Sejvar, André Miguel Japiassú.

**Data curation:** Hugo Boechat Andrade, Margareth Catoia Varela, Grazielle Viana Ramos, Aline Ramos da Silva, André Miguel Japiassú.

**Formal analysis:** Hugo Boechat Andrade, Ivan Rocha Ferreira da Silva, José Henrique Mello-Neto, Mayara Secco Torres da Silva, Margareth Catoia Varela, Aline Ramos da Silva, André Miguel Japiassú.

**Funding acquisition:** Ermias D. Belay, José Cerbino-Neto.

**Investigation:** Hugo Boechat Andrade, Ivan Rocha Ferreira da Silva, Justin Lee Sim, José Henrique Mello-Neto, Pedro Henrique Nascimento Theodoro, Mayara Secco Torres da Silva, Margareth Catoia Varela, Grazielle Viana Ramos, André Miguel Japiassú.

**Methodology:** Hugo Boechat Andrade, Fernando Augusto Bozza, José Cerbino-Neto.

**Project administration:** Hugo Boechat Andrade, José Cerbino-Neto, André Miguel Japiassú.

**Resources:** Hugo Boechat Andrade, José Henrique Mello-Neto, Grazielle Viana Ramos, Ermias D. Belay, José Cerbino-Neto.

**Software:** Hugo Boechat Andrade, Margareth Catoia Varela.

**Supervision:** Fernando Augusto Bozza, Jesus Soares, Ermias D. Belay, André Miguel Japiassú.

**Validation:** Ivan Rocha Ferreira da Silva, Justin Lee Sim, José Henrique Mello-Neto, Pedro Henrique Nascimento Theodoro, Aline Ramos da Silva.

**Visualization:** Hugo Boechat Andrade, José Cerbino-Neto, André Miguel Japiassú.

**Writing – original draft:** Hugo Boechat Andrade.

**Writing – review & editing:** Hugo Boechat Andrade, Ivan Rocha Ferreira da Silva, Fernando Augusto Bozza, Jesus Soares, Ermias D. Belay, James J. Sejvar, José Cerbino-Neto, André Miguel Japiassú.

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
