## [Decision Letter · Decision Letter 0]

19 Oct 2021

PONE-D-21-19498Nervous system infection in the intensive care unit: development and validation of a multi-parameter diagnostic prediction tool to identify suspected patientsPLOS ONE

Dear Dr. Andrade,

Thank you for submitting your manuscript to PLOS ONE. After careful consideration, we feel that it has merit but does not fully meet PLOS ONE’s publication criteria as it currently stands. Therefore, we invite you to submit a revised version of the manuscript that addresses the points raised during the review process.

We look forward to receiving your revised manuscript.

Kind regards,

Andrea Cortegiani, M.D.

Academic Editor

PLOS ONE

Journal Requirements:

Additional Editor Comments (if provided):

Two reviewers experts in the field underlined the quality of the paper and proposed some points to be improved. I agree on these points that should be addressed.

Reviewers' comments:

Reviewer's Responses to Questions

**Comments to the Author**

1. Is the manuscript technically sound, and do the data support the conclusions?

Reviewer #1: Partly

Reviewer #2: Yes

2. Has the statistical analysis been performed appropriately and rigorously? 

Reviewer #1: I Don't Know

Reviewer #2: Yes

3. Have the authors made all data underlying the findings in their manuscript fully available?

Reviewer #1: Yes

Reviewer #2: Yes

4. Is the manuscript presented in an intelligible fashion and written in standard English?

Reviewer #1: Yes

Reviewer #2: Yes

5. Review Comments to the Author

Reviewer #1: The paper is well written. The authors specified a model to indicate the diagnosis of central nervous system infections. Their ultimate model included AIDS/HIV, Age, CSF WBC >2 cells/mm3, encephalopathy, fever, focal neurologic deficit, GCS <14 (points), seizures as the parameters to establish CNS diagnosis in suspected ICU patients. Frankly to say, I was not surprised with these results considering our daily medical practices. Anyway, it may contribute to the readers. On the other hand, it will be better if an expert statistician checks advanced mathematics in the paper.

There are some minor points: Please revise NSI and replace it with central nervous system infections. The reference of erdem et al mentions all CNS infections, HIV positives are a subgroup in the paper and should be revised accordingly.

Reviewer #2: In the article “Nervous system infection in the intensive care unit: development and validation of a multi-parameter diagnostic prediction tool to identify suspected patients” the authors present an interesting tool that could aid physicians screening patients for nervous system infections. The article is well presented, clear in the content and appropriate in the form. Results are clearly presented and there are not overstatements in the discussion. The study limitations and conclusions are adequate, reporting the need for this tool to be further validated in different contexts from the ones of this study. I do not have major concerns regarding the publication of this article.

As a minor concern I noticed the absence of a protocol registration which might have been appropriate.

Other comments:

Line 144: please rephrase as the meaning of the sentence is unclear.

Line 195: please rephrase as it follows: “The variables required for calculating theVC2 were collected and crosschecked by two of the authors (IRFDS and JLS)”

6. PLOS authors have the option to publish the peer review history of their article (what does this mean?). If published, this will include your full peer review and any attached files.

Reviewer #1: **Yes: **Hakan Erdem

Reviewer #2: No

---

## [Author Response · Author response to Decision Letter 0]

21 Oct 2021

PONE-D-21-19498

Nervous system infection in the intensive care unit: development and validation of a multi-parameter diagnostic prediction tool to identify suspected patients

PLOS ONE

New title after review: “Central nervous system infection in the intensive care unit: development and validation of a multi-parameter diagnostic prediction tool to identify suspected patients”

Response to Reviewers

“Dear Dr. Andrade,

Thank you for submitting your manuscript to PLOS ONE. After careful consideration, we feel that it has merit but does not fully meet PLOS ONE’s publication criteria as it currently stands. Therefore, we invite you to submit a revised version of the manuscript that addresses the points raised during the review process.”

R: Thank you all for the attention and consideration. My answers are highlighted in blue.

"If applicable, we recommend that you deposit your laboratory protocols in protocols.io

to enhance the reproducibility of your results."

R: Thank you for the information. There are no laboratory protocol for this study, so it's not applicable.

“Additional Editor Comments (if provided):

Two reviewers experts in the field underlined the quality of the paper and proposed some points to be improved. I agree on these points that should be addressed.”

R: Thank you. We addressed the points to improve it. It is an honor to publish once again with such an internationally respected medical journal.

“5. Review Comments to the Author

Reviewer #1: The paper is well written. The authors specified a model to indicate the diagnosis of central nervous system infections. Their ultimate model included AIDS/HIV, Age, CSF WBC >2 cells/mm3, encephalopathy, fever, focal neurologic deficit, GCS <14 (points), seizures as the parameters to establish CNS diagnosis in suspected ICU patients. Frankly to say, I was not surprised with these results considering our daily medical practices. Anyway, it may contribute to the readers.

R: Thank you very much for your time. Yes, our intention was to build a tool that could help general practice physicians screening patients for central nervous system infections.

“On the other hand, it will be better if an expert statistician checks advanced mathematics in the paper.”

R: Yes, good advice. Thank you. The advanced mathematics were really complex, so they were assessed by expert statisticians from our institutions before submission.

There are some minor points: Please revise NSI and replace it with central nervous system infections. The reference of erdem et al mentions all CNS infections, HIV positives are a subgroup in the paper and should be revised accordingly.”

R: We changed Nervous System Infection (NSI) for Central Nervous System Infection (CNSI) as recommended. Therefore, this led to the title of the article being changed to “Central nervous system infection in the intensive care unit: development and validation of a multi-parameter diagnostic prediction tool to identify suspected patients”.

We also revised the text for the reference of Erdem et Al (lines 342-347) as follows: “The microbiological profile is compatible with the current international literature: in a multicenter international study to understand the burden of community acquired CNSI, Erdem et al. showed that the most frequent pathogens were Streptococcus pneumoniae (n=206; 8%) and Mycobacterium tuberculosis (n=152; 5.9%). Cryptococci were the leading pathogens in the subgroup of HIV-positive individuals. Ninety-six (8.9%) patients of INI's sample presented with clinical features of a subacute disease, suggestive of tuberculosis or neurosyphilis [31].”

“Reviewer #2: In the article “Nervous system infection in the intensive care unit: development and validation of a multi-parameter diagnostic prediction tool to identify suspected patients” the authors present an interesting tool that could aid physicians screening patients for nervous system infections. The article is well presented, clear in the content and appropriate in the form. Results are clearly presented and there are not overstatements in the discussion. The study limitations and conclusions are adequate, reporting the need for this tool to be further validated in different contexts from the ones of this study. I do not have major concerns regarding the publication of this article.

As a minor concern I noticed the absence of a protocol registration which might have been appropriate.”

R: Thank you for your review. As an observational study with no interventions and no systematic reviews, we didn’t register the protocol. However, with your comment, we have only recently discovered that there is a possibility to register observational studies on the ClinicalTrials.gov website. Thank you for the advice. Next time we will proceed as recommended.

“Other comments:

Line 144: please rephrase as the meaning of the sentence is unclear.”

R: We agree. We deleted part of it, as it was confusing. The new phrase is as follows: “Two physicians (HBA and JHN) independently reviewed the medical records. The diagnosis of CNSI was considered if it met at least two of…”.

“Line 195: please rephrase as it follows: “The variables required for calculating theVC2 were collected and crosschecked by two of the authors (IRFDS and JLS)”’

R: Done. Rephrased.

Sincerely,

Hugo Boechat Andrade, MD, MSc

Corresponding Author

---

## [Editor Report · Decision Letter 1]

12 Nov 2021

Central nervous system infection in the intensive care unit: development and validation of a multi-parameter diagnostic prediction tool to identify suspected patients

PONE-D-21-19498R1

Dear Dr. Andrade,

We’re pleased to inform you that your manuscript has been judged scientifically suitable for publication and will be formally accepted for publication once it meets all outstanding technical requirements.

Kind regards,

Andrea Cortegiani, M.D.

Academic Editor

PLOS ONE
---

## [Editor Report · Acceptance letter]

16 Nov 2021

PONE-D-21-19498R1 

Central nervous system infection in the intensive care unit: development and validation of a multi-parameter diagnostic prediction tool to identify suspected patients 

Dear Dr. Andrade:

I'm pleased to inform you that your manuscript has been deemed suitable for publication in PLOS ONE. Congratulations! Your manuscript is now with our production department. 

Kind regards, 

on behalf of

Dr. Andrea Cortegiani 

Academic Editor

PLOS ONE